# Hermeneutic-Phenomenological Interpretation of Coronavirus Experiences, Their Meanings, and the Prospects of Young Finns in Education and the Labor Market in Lapland

Helena Marketta Helve 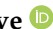

Faculty of Social Sciences, Tampere University, 33100 Tampere, Finland; helena.helve@tuni.fi; Tel.: +358-405464493

**Abstract:** In this paper I reflect on the methodological concepts of youth research, utilizing a hermeneutic-phenomenological approach to the interpretation of interview data from young adults who have been on short-term work or distant education at Finnish ski resorts in Lapland during the coronacirus pandemic. The study received background from a previous study "From higher education to working life: Work values of young Finns in changing labor markets". I try to distance myself from this research by interpreting young people's coronavirus experiences and future perspectives hermeneutic-phenomenologically. In the spring of 2021, I interviewed a total of ten (5 women and 5 men) young people aged 19 to 27 I met at the ski resorts. Interviews on young people's coronavirus experiences and their implications for the transitions from education to employment and future orientations were semi-structured, partly discussions of topics related to education, work and transition to adulthood combined with young people's COVID-19 experiences and their implications. In the interviews, young people combined their previous life experiences and perceptions of the world with the coronavirus experiences. The coronavirus experiences of young people were situational. The study analyzes the individual experiences of young people with the COVID-19 pandemic, describing them with own youth spoken language, and interpreting the essential contents of the meanings hermeneutic-phenomenologically. The COVID-19 interpretations of young people had positive and negative meanings to their transitions in education and the labor market. The basic themes that cut across the entire material were: (1) The small impact of the pandemic on the young person's own life. (2) The uncertainty of life and uncertain future and (3) the experienced loneliness, which can provide for youth to confront their true selves. The implications of these results are discussed in the article, which also critically considers the applicability of the hermeneutic-phenomenological research, and discusses about ethical points of the study of young people in exceptional contexts.

**Keywords:** Youth; COVID-19 experiences; transitions; future orientation; hermeneutic-phenomenological research

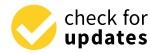

## 1. Introduction

Well, here last year in the spring, I lost a work position in a café due to the situation, but I was lucky to find another job in a store, but in the summer and pretty much in the fall, I did what was planned, and they weren't affected. For work, everything was all right in the end. . . . Social life has been cracking; you can see that people don't care to come visit because there are a lot of tourists from the south, and the number of visitors has been decreasing, and now I haven't seen friends as much; being in a group is in the past for the most part. And, like, evenings out and so on are clearly decreasing in my life too.

(A 21-year-old female student applying for art education, interview in Lapland, April 2021, Maija, 21-year-old female student.)

The COVID-19 coronavirus pandemic is an ongoing global pandemic. The World Health Organization (WHO) declared a Public Health Emergency of International Concern

on 30 January 2020, and a pandemic on 11 March 2020. According to UNESCO (2020) [1], by the end of April 2020, schools and higher education institutions were closed in 178 countries, affecting roughly 1.3 billion learners worldwide. Lockdowns, restrictions on movement, the disruption of routines, physical distancing, the curtailment of social interactions, and the deprivation of traditional learning methods have led to increased stress, anxiety, and mental health concerns for young people worldwide (UNESCO, 2020) [2]. In Finland, restricting movement on 25 March 2020 was a drastic measure from the Finnish government of Prime Minister Sanna Marin (Emergency Law 118; see more Scheinin, 2020) [3]. Only necessary commutes were allowed. However, the units of early childhood education and the preschool education associated with them were kept in operation. In this way, for example, the employment of the children's parents was secured. The facilities of schools, educational institutions, universities, and polytechnics, as well as civic colleges and other free educational institutions, were closed, and in-person teaching was suspended (the regulations in Finland entered into force on 18 March 2020). Instead of in-person teaching, the teaching of universities, polytechnics, vocational and upper secondary education, and basic education was organized in alternative ways such as distance learning and various digital learning environments. In Finland, universities stopped providing onsite learning on March 18. (The COVID-19 pandemic forced most universities to switch from in-person to remote teaching from May 2020 to May 2021). Public gatherings were limited to ten people. (Museums, theaters, the National Opera, youth and culture houses, libraries, the National Archives' customer and research hall services, hobby facilities and venues, swimming pools and other sports facilities, club facilities, day care activities for the elderly, rehabilitative work, and visits to housing services for the elderly and other at-risk groups were prohibited). Public sector workers switched to telecommuting to the best of their ability. Those over the age of 70 were required to remain in quarantine-like conditions, except for MPs, state leadership, and municipal trustees (OKM, 2020 [4]; my own diary entries from March 2020 to April 2021).

In 2020, Finnish Lapland lost 95 percent of its business. Many companies in tourism were impacted by the coronavirus and the different restrictions set on travel and tourism as a result. This meant a loss in demand for their services and thus a decrease in their revenue. Many companies relied heavily on foreign tourism, and the lack of it caused financial losses for them. Not only were professional staff being lost, but also the future of the companies was uncertain. Additionally, different measures were taken to reduce the financial impact of the coronavirus. (These measures included, for instance, government support, reductions in staff, and increased borrowing (Rekilä, 2021) [5]). The spring season of Lapland's winter tourism is usually busiest during the ski holiday weeks. Lapland attracted domestic tourists already in the spring season of 2021. (In December 2021, European tourists arrived in Lapland, and the record for the number of overnight stays at the northern ski resorts was broken, which was higher than before the COVID pandemic over Christmas of 2019 [6] (https://www.stat.fi/til/matk/index.html; Statistics Finland, https://visitory.io/en/lapland/) (accessed on 17 March 2022)).

This research is a continuation of a study from ten years earlier about young, short-term employees at the ski resorts of Lapland [7,8]. (The changing lifestyles and values of the young persistently temporarily unemployed in the different labor markets of Finland 2008–2011 (WORK-Preca) funded by the Academy of Finland [9]). The hypothetical point of departure for this study was that short-term and temporary employment is changing the identity, future expectations, work attitudes, values, and worldviews of young people. (The in-depth narrative interviews and ethnographic observations made over an eight-week period were gathered in 2009–2010 among twenty (*n* = 20) young people from 17 to 28 years old working temporarily in tourism in Lapland). Many of them saw the future as cyclic, from the winter season to the Easter season, from the summer season to the autumn season, and the Christmas season. During the summer, when many of them were unemployed, they traveled and lived with their parents, who helped them if needed. They had formed

their own individual concepts of time. This was also part of their identity construction and plurality of identities [10–12].

In the social discussion related to COVID-19 in Finland, concern has been highlighted about the effects of the pandemic on young people and young adults. They have even been referred to as the lost generation. There is particular concern about their weak labor market position and their unequal opportunities to succeed in distance learning. The social conversation about the everyday life of young adults during the pandemic period is often left without listening to the young themselves. In the Finnish ALL-YOUTH research project, an effort has been made to listen to the stories of young adults about what it is like to live as a young person in the COVID-19 era (https://www.allyyouthstn.fi/en/all-youth-2/) (accessed on 17 March 2022). This article will describe research on the pandemic experiences of young people in Lapland.

## 2. Youth Studies of the Corona Effects

The effects of the epidemic on young people have been studied in Finland according to, for example, socioeconomic status, health, ethnicity, party affiliation, sexual orientation, and gender [13–16] (Lahtinen and Salasuo, 2020). The literature on COVID-19 youth research gives a rather vulnerable picture of the impacts of the pandemic on children and young people. In health-related studies, it has been ascertained that the psychological impact of the COVID-19 pandemic has increased anxiety and stress associated with lack of access to normal activities of daily living and financial concerns due to the economic consequences of "lockdown" [17] (cf. Bentall, 2020). In the context of COVID-19, social isolation and physical distancing appear to affect children and young people via worsening mental health. The research on school children in the era of COVID-19 has evidence of many negative consequences of school closure, e.g., from social isolation to the disruption of routines and a lack of structure in life, uncertainty about the future, and reduced levels of enjoyable activities and physical activity. Young people may worry about uncertainty about their future, which makes them fearful, and they may have stomach and headaches related to the intense stress [18]. These are emotional responses to the circumstances of COVID-19. In the UK Youth Survey (2020) [19] about the impact of COVID-19 on young people and the youth sector, the focus was on young people's experiences in education and in employment. The UK Youth Survey 2020 had evidence of increased mental health or wellbeing issues, increased loneliness and isolation, a lack of safe spaces, including not being able to access youth clubs and services and a lack of space at home, challenging family relationships, and a lack of trusted relationships, which increased the use of social media with online pressures and the possibility of engaging in harmful gangs, substance misuse, carrying weapons or other harmful practices, and a higher risk for sexual exploitation or grooming. However, there might be a danger of too much medicalization of all the reactions of young people.

ILO has analyzed in the Report of Youth Employment in Times of COVID [20] that young people have been hit hard in both economic and social terms. The lower-qualified young people already are twice as likely to become NEET (Not in Education, Employment, or Training) youth. Young people are also overrepresented in the gig economy, on temporary and zero-hour contracts, which means that they are less protected by labor laws and therefore more vulnerable. The ILO Report 2021 highlights that young women have been more impacted by the crisis in terms of job losses and reduced incomes than young men, the labor market situation and opportunities for many young people were already precarious and insecure before COVID. These findings explain the concept of a "COVID-19 or lockdown generation" that has slowly emerged in the media to describe how young people may be scarred for decades to come, in terms of the labor market and mental health outcomes.

The unemployment rates of all recent Finnish graduates from vocational schools and university education during the corona period seem to have remained quite small. The Finnish statistics on the employment of young adults show that the employment rate of 25–34-year-olds in 2020 is higher than in the corresponding period in 2019. However, the

wages of young people have remained lower than before the pandemic because young people have more short-term temporary employment relationships, furloughs, and dismissals than others. This can bring pressure to a transitional phase of life, where many young people are thinking about their adulthood and ways of living [21].

## 3. Youth Research in Exceptional Contexts

While I was in isolation in spring 2020 during the COVID-19 pandemic in Lapland, my intention was to collect interview material about the coronavirus period experiences of seasonal workers under 29 years at Lappish ski resorts. I was interested in the coronavirus experiences of young adults and their implications for their future planning and horizons [22]. (The uncertainty of arranging and conducting interviews during the coronavirus period removed the basis from my previous experience of conducting interviews in Lapland. Plans for data collection schedules, for example, were unsuccessful due to the constraints imposed by the coronavirus. I had already twice planned interviews with young people in Lapland, but due to the coronavirus period, the tourist centers were closed, and I did not meet young seasonal workers for my research).

The coronavirus period brought new challenges for me as a researcher, who had been conducting youth research for more than 45 years, requiring the ability to be flexible and cope with the conditions, such as the use of masks, safety intervals, and interviews partly outside in the frost. I had to focus on the difficult accessibility of young people during the coronavirus period. Exceptional circumstances call for resilience and new solutions in reaching out to young interviewees.

This article reflects youth coronavirus experiences, their meanings, and the prospects of young people in education and the labor market in Lapland from the hermeneutic-phenomenological framework of interpretation. The young people in this study are from 19 to 27 years old.

During the pandemic, the duties of young people at ski resorts were mostly performed alone. During the day, I went to the ski slopes, ski and hiking centers, ski ticket sales, sports shops, and open dining places to find young people to interview. I did ten ($n = 10$) interviews (five young women and five men) from March to April 2021. The interviews were face-to-face, and some were conducted outdoors. I first talked about this study, COVID-19 pandemic-related themes, and our Finnish COVID-19 research team, which includes this research on young people's coronavirus experiences in Lapland. (The group of youth researchers are from different Finnish Universities. See All Youth https://www.allyouthstn.fi/in-english/ (accessed 17 March 2022)) [23].

In this study, young interviewees anonymously represent young seasonal workers in Lappish tourist centers. The young people came from middle-class families. The parents of several were entrepreneurs. Several had spent their childhood holidays in Lapland with their families. Three even lived in their own cottage in Lapland. Compared to my previous study, these young people had better living conditions precisely because of the coronavirus; for example, employers did not have large common accommodations for seasonal workers, but they were placed in private housing.

Most of the interviewees were in their twenties. Everyone had a high school diploma, and a couple of them were applying for art education in the spring. The basic education degrees were in tourism; for example, business economy (two), business administration, and restaurant economy. One of the interviewed young people was a first-year student at university. She was studying engineering by distance in Lapland with her friend. The seasonal work at the ski resorts included the refurbishment, sale, and rental of sports shop equipment; the sale of sportswear and ski passes; the teaching of cross-country and downhill skiing and snowboarding; guidance on nature hikes; photography; and updating the Lapland-themed digital platforms.

When I met young people to recruit them for an interview, I explained that the purpose of the study was to find out their personal coronavirus experiences. The interviews were in the form of a discussion. Talking about the COVID-19 research before the interviews created

an open atmosphere for later interviews and gave the interviewees a chance to think about the topic in advance. Suitable times and places for the interviews for the young people were agreed upon, as well as the recording of the interviews. Those ten young people I met in Lapland at random wanted to participate in this larger study of young people in exceptional COVID-19 pandemic circumstances. (I think that openness and reliability in the interview situation gave me the opportunity to get to know the interviewee in advance before the interview situation itself). (See Figure 1).

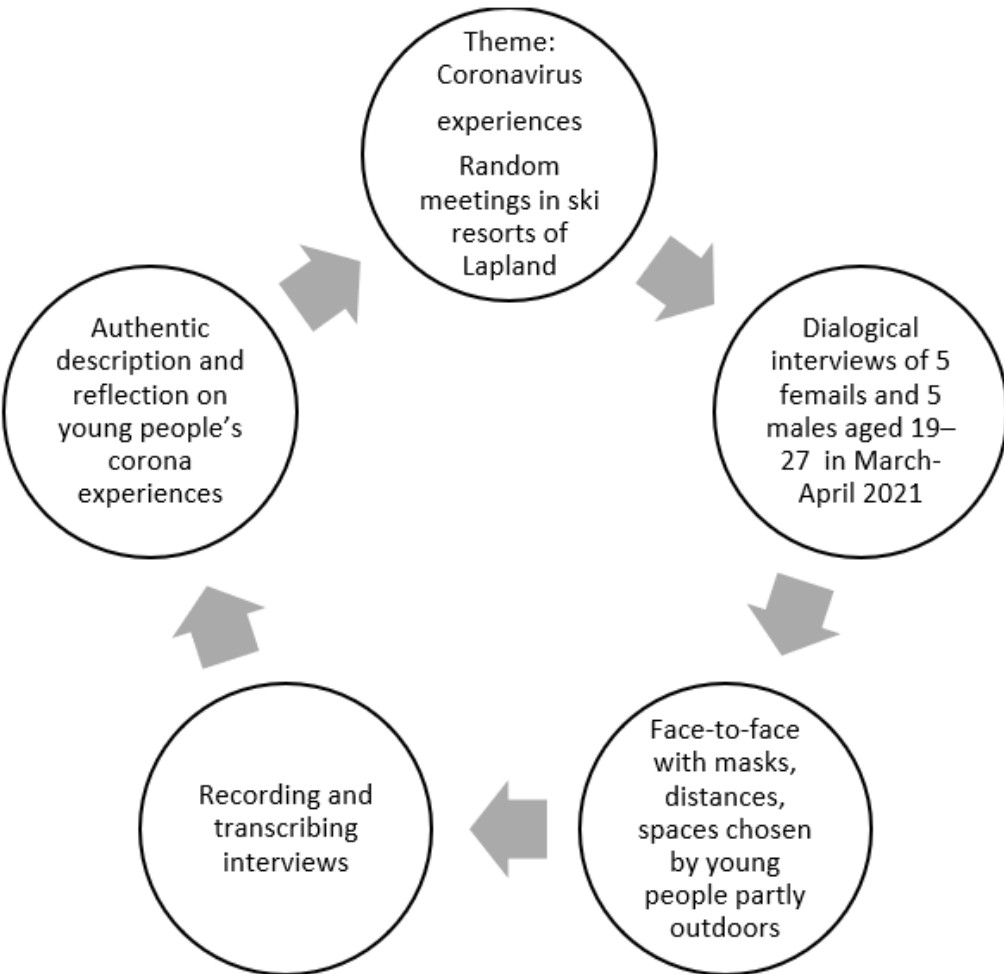

**Figure 1.** Collection of research material.

At all moments of the interview, the interviewee and the researcher had masks on their faces. I also had hand sanitizer with me. Some of the interviews were conducted indoors with safety intervals. Outdoor interviews were conducted at the outdoor tables and seats of the ski and hiking centers, also at safety intervals. The outdoor interviews were partly disrupted by outdoor sounds, which brought tension about how difficult it would be to transcribe the interviews. Fortunately, this was not the case. On average, an interview conversation lasted from half an hour to an hour. The main topics of discussion were related to the effects of COVID-19 on the young person's own life. (In part, the themes were the same as those of the other researchers in the Finnish COVID-19 research group. We had discussed these themes in our online team meetings. The theme of the future was also in my previous study [24]. This time, however, the focus of the study was on the interviewees' own experiences of the coronavirus era).

The research ethics emphasized the voluntary willingness to participate in the study, which was clear in the random reach of young people. The aim of making the study as open as possible was to arouse interest in it, increase the reliability of the research, and protect the

privacy and anonymity of the subjects. The ethical questions in the study were the random recruitment of the interviewees and the prior information presented about the study. Interview situations were not expected to pose significant risks, harm, or inconvenience to the young persons under study, those close to them, or other people, nor to the anonymity of the interview situations. This also applies to the subjects' right to self-determination and respect for volunteering, as well as trust in the researcher and science [25].

## 4. Hermeneutic-Phenomenological Research Framework

This qualitative study has a philosophical approach not bound by the structured stages of a method. However, there are sources that address the practical aspects of how to carry out a phenomenological study [26–34]. The research question impacts the whole research process, for which Gadamer (1960) [26] used the concept of gaining insight to gather research data. This is possible through open dialogue between the researcher and the subjects of the research. (Gaining an understanding through interview dialogue, and with the interview transcript text and its analysis, listening to the recordings of the interviews once again, I wrote my own field notes next to the transcript text of the interviews, which describe the context and emotions from the recordings [27,33,34]). In this article, *hermeneutics* emphasizes how young people interpret the coronavirus phenomenon and its meanings for them, and *phenomenology* emphasizes the coronavirus experiences of young people and the formation of an understanding of it.

The study investigates young people's individual experiences of the COVID-19 pandemic, describing them and interpreting the essential contents of their meanings. The hypothetical idea of the hermeneutic-phenomenological study is that young people's private interpretations could be combined into themes that form structures of meaning according to a hermeneutic approach and help to find the general structures of youth COVID-19 experiences as young people have generally experienced them. The default is that the youth's freely spoken language is shared between interviewees [27,35]. (Youth spoken language is shared between young people. According to Gadammer (2006) [35], language opens access to meaning and understanding of the world). Understanding the whole text was the starting point for the analysis because the meaning of the whole text affects the understanding of every other part of the text. Texts were constructed from the transcription of interviews and my diary and field notes. I tried to keep my mind open and reflect on my own influences while reading transcript texts and listening to the audio recordings of the interviews. This helped in the preliminary interpretation of the texts and the coding of the interview texts. (I used the NVivo12 software program) [36]. The problem in this English article is the translation from the Finnish spoken language to English. The translation might lose a genuine interpretation of the young person's meaning.

As I read the texts, I tried to find the main meanings and obtain a more detailed understanding of the meanings of the COVID-19 pandemic. It was challenging to find the interviewee's view, the horizon, referring to the meanings given to the coronavirus experience by the young person in his or her own words or expressions. (See Figure 2.)

The synthesis and theme development represent my horizons, forming abstractions based on my research knowledge about the studies of worldview formation, social capital, and identity horizons [8,37,38]. I formed the illustration of the synthesis by grouping the subthemes with the help of the Mindomo mind map (https://en.wikipedia.org/wiki/Mindomo [39]). The literature from the whole material was linked to the identified themes and subthemes. Based on the themes, the subthemes, and their interrelationships, I reconstructed young people's coronavirus experiences into a story in their own words (so-called first-degree constructs) to illuminate phenomena and highlight key observations from the interview data [27,33,34,40,41].

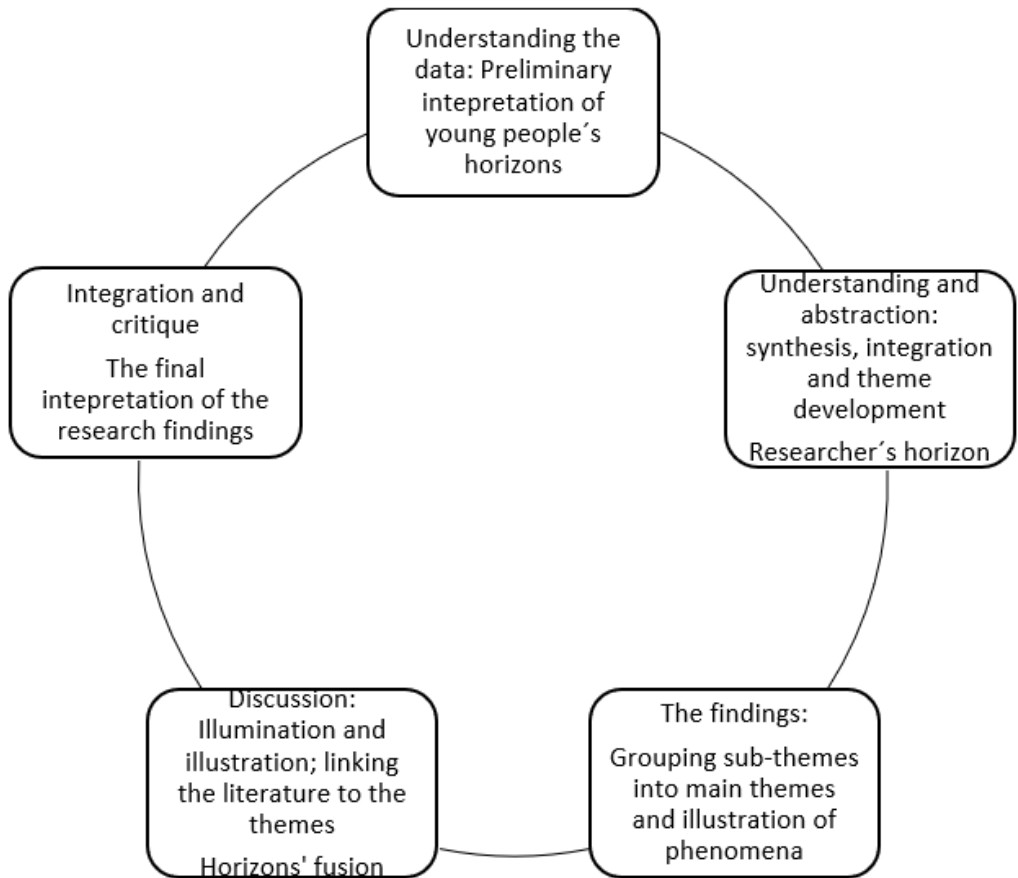

**Figure 2.** Intepretation of findings in the Hermeneutic Circle.

In hermeneutic-phenomenological research, methodological consistency required openness and sensitivity when listening to the interview of each participant to ensure that their perspectives (horizons) were clearly noticed and presented as authentically as possible. (Methodological consistency should be reflected throughout the research process in a nonlinear manner as the researcher moves back and forth between the frame of reference to ensure consistency between the research question, the literature, data collection, and analysis [34,42]). Providing direct quotes from interview transcripts helped to validate the data and to assess the reliability of the research. However, the interpretation of the data changes over time. According to Gadamer's philosophy, the horizons of the researcher and the research participants are changing, and therefore, a definitive interpretation cannot be reached. (The translation of the youth's spoken language may change the interpretation. Youth language is taken to refer to all patterns of language used in the social age of adolescence in one's own mother tongue).

During the analysis and writing phases of the study, I entered the material into NVivo 12 (https://en.wikipedia.org/wiki/NVivo). In the analysis of the material, I went through the interviews many times, checking forward and backward, noting whose voices were missing, and trying to find similarities and differences in the coronavirus stories. Because of the coronavirus, interviews were conducted in different places, and this can also affect the young people's reactions to the interview situation.

As the researcher, I had to carefully identify and reflect on my own position and thoughts on the phenomenon under study. I followed the news of the pandemic daily and tried to describe my own experiences and realize how they might potentially affect my analysis. I also sought different insights into the conditions of young interviewees in Lapland so that I could understand and interpret their experiences of the coronavirus. It is not easy to capture and explore the meanings that young people assign to their

experiences in the subjective and reflective process of interpretation that the researcher enters during the research process. The findings of this study are the themes and stories of young people [34,41]. The quotations of young people are verbal explanations and information, which are linked in the analysis to the themes and literature. Due to the number of codes, I used the computer program NVivo12 to assist the data organization and form theme categories. The subcategories represent my horizons as the researcher. These were generated using my theoretical and personal knowledge of adolescence and young people. At the abstraction level, an illustration of how subthemes evolved to the final themes is presented in the findings in the next chapter.

## 5. Research Findings

### 5.1. COVID-19 Experiences

Horizons of Young People

In this article, the quotations of the private speaker and his or her translated transcript are distinguished by pseudonym name and by age. Also, the place where the person worked is mentioned. The individual interview texts together form a narrative for analysis. The first analysis gives a picture of young people's horizons. They spoke about their COVID-19 experiences during their time spent in Lapland, their future plans, their emotions, their families and friends, their hobbies, and their fears and dreams. Their stories give a very wide insight into their pandemic experiences.

> I came for the first time last season. I started in January. Was just looking for work here somewhere, wanted to go downhill skiing, I was in Helsinki. Then I just applied and decided to come this season. Now I have been here since November. ... I got COVID here last year.
>
> (Maija, 21-year-old in Sport Shop.)

> ... Yeah. When we went, like, earlier. Around the time when we closed, in the apartment where I live, we all had COVID. I think there were 10 of us. ... One day I had a very high fever and felt sick, but the next day I went to the arctic hills to cross-country ski. I didn't have anything more than that. ... I have been in quarantine already once this season. So, it has affected my life quite a lot, has changed how we do things normally ... that you think about parents or grandparents. I'm not scared about it personally in any way, but I think about older people, and for them it's a bit scary for sure.
>
> (Lauri, 26-year-old in Ski Rental Company.)

All interviewed young people had experiences with COVID-19 in their life. Some had had it and been in coronavirus quarantine; some had experienced the death of someone; all had remote connections to their families and friends. They experienced COVID-19 as a serious issue, and they were following the government's recommendations. The impacts of the experiences during the coronavirus pandemic in Lapland were both negative and positive. The ability to exercise outdoors was experienced as a positive impact on hobbies and leisure activities. Some attested that COVID-19 had not affected their life directly.

> Well, I would say that it hasn't really affected my life directly. I have a lot of hobbies that I can still do. A lot of the hobbies happen outside, like skiing and downhill skiing and other things like that. And in downhill skiing, it's mostly in nature rather than the hills that are being maintained for it. And here, the number of friends is small anyway.
>
> (Ritva, 22-year-old in Ski Shop.)

> ... So, although it's confusing, sometimes even feels wrong that life is going so well for me when others are struggling, and in cities it's probably a completely different thing, but here it seems like life is going on, and then reading the news and being like, "oh no". ... but here you somehow feel that life goes on, and this gives opportunities for extensive action here; it also allows me to work on

a flexible schedule ... I've been able to do a lot of work remotely even before the coronavirus. It has been very important to allow yourself that freedom. Not so tied to time and place. That of course some things have to be taken care of carefully, and sometimes you have to be in a certain place, but in general.

(Pekka, 27-year-old in Nature Center.)

Everyone had experienced some negative impacts on their social life. For some, the COVID-19 pandemic did not have a very negative impact on their relationships with family members because they were satisfied with their connections to them through social media. There were young people who said they were introverts and happy about not seeing many people around.

Clearly, many young people, especially students, certainly have quite a lot of pressure and difficulty to complete school, and so a lot of content is shared through ... that is gracious to yourself and try to take it easy at times.

(Erkki, 21-year-old in Sport Shop.)

I'm also thinking about work stuff ... that I've somehow built everything in my own life anyway, in such a way that it isn't the end of the world, even if the work goes away. Or I have a rather optimistic way to think that everything will always be in order anyway. Well, now it's going to go deep, but then there's a lot of contemplation, so if it's the end of the world tomorrow or something happens, you wouldn't die, if there's something I must regret, not really.

(Pauli, 26-year-old in Ski and Hiking Centre.)

The interviewees felt that even though they were working in ski resorts, they lived in their own "bubble." Their social life was limited because their parents, grandparents, and friends lived in other places. However, they had connections to them through social media. Some of the Lappish youth themselves also had their own platforms to share ideas and photos. Some even saw social media as a career possibility.

... on social media ... Well, not very active. I have all the social medias, but I don't really use them. ... Well yeah, it's been less active in the past few months, but sometimes something, checking in on others and updating my own things. Usually something related to the outdoors or being outside, I put a lot about what I do here ... go in like seasons, like the last couple weeks I've been very quiet on social media, and at some point, there will be more content there again, but it's also like, this is a cool picture, I should post this here.

(Laura, 20-year-old in Ski Rental.)

It's not a thing I must do, or that it would be a routine somehow, more just when I feel like it. ... For example, making and producing a music video is quite a big thing, so in that industry, there should be a lot of work, especially if you know how to do it. Well, in the music industry, you just got to take your own path and know your thing, or pretty much on any career path, if you know how to do it well, there should be work, but now due to COVID, a lot more is done on social media, and it's used much more, and for example, all arts are followed very much there ...

(Erkki, 21-year-old in Sport Corner.)

Well, yeah, through social media it has come through, at least to me, or it seems to me that the influencers have encouraged people and somehow help the people in this situation through social media.

(Emma, 21-year-old in Sport Shop.)

Social media was a way to connect young people to their families and friends. One of the most relevant characteristics of social media platforms in this COVID-19 pandemic has been the rapid dissemination of protocols at the regional, national, and international levels.

The platforms are among the most widely used sources of information in the world, and the easy and inexpensive access to the internet and many registered users on these platforms make them one of the easiest and most effective ways to disseminate information.

### 5.1.1. Meanings for Coronavirus
#### 5.1.1.1. Concerns

The young people did not want to risk the health of others. The greatest concern was about grandparents, even if they were vaccinated. They were also concerned about people nearby if they fell sick or if work stopped and they lost their living conditions. A seasonal worker at a restaurant had experienced three weeks when people were not able to eat inside the restaurant. During that time, they had organized takeaway food, and luckily, none of the workers needed to be laid off. Some of the young people in short-term jobs in Lapland expressed specific financial concerns. These included frustration and anxiety regarding reduced income and fewer career opportunities in tourism. Reduced income combined with fears of unemployment amplified some financial stress [20] (cf. Barford et al., 2021).

> . . . Like, if my own finances are safe, it's easier to breathe. . . . For example, my parents live in the south, so I always have to think if I can now go there, and I always take a COVID test before going anywhere further. There's always a bit more organizing to be able to see friends. We are very close with our colleagues during free time, because we are in the same groups anyway. But if you go outside the bubble, you have to think about these restrictions and so on, so that you don't risk others' health. . . . Well, I have noticed that, for example, parents, even though they live in Joensuu, we can keep in touch remotely, and we have done Zoom meetings with the family. Even though we can't always physically see each other, remotely works. For example, my grandparents make me worried. They have received the vaccine, but my grandfather. . . . He has had cancer and other things. . . . So I'm worried about that, and about my own career, that the work can end at any time.
>
> (Ritva, 22-year-old in Sport Shop.)

> . . . A friend's brother had COVID, and sadly my friend's grandmother died from COVID, but her health was also very bad anyway, but it's like . . . I don't feel I've had a lot of contact with COVID. They've all been quite distant, for which I feel lucky. . . . It hasn't affected my long-term goals.
>
> (Emma, 21-year-old in Sport Shop.)

COVID-19 made young people think about the health of their loved ones. They followed the given pandemic instructions. In a study by Appleby et al. [43], female students at different universities were more likely than males to report following recommendations and adapting their lifestyles. This may be partly due to increased anxiety among females, which has also been reported in a Finnish study of university students [44].

#### 5.1.1.2. Loneliness

Loneliness during COVID-19 emerged as one of the main themes in the interviews. It may pose a serious threat to youths' survival, health, and longevity [17,18,45]. In this study, there were also positive meanings about how loneliness could be essential for happiness and flourishing. Young people may have a natural tendency to avoid anything uncomfortable, but many found that during COVID-19, they had time to think about their life and they discovered self-understanding and life projects of self-transcendence.

> I could have studied for five years without thinking if this is the correct field of work for me. . . . And now I have spent a lot of time on it, and I have been thinking if this is the correct career and this field of study is the one I like. . . . So, it's, like, something that I can partly be happy about. So at least now, if I am here in Lapland, I would be happy to continue my studies, so I know my career

is correct, and it's a career that somehow COVID has shown that I can make a change with in the future . . . that I could work on a master's degree maybe somewhere else and then graduate from somewhere else, because . . . Hopefully abroad. But of course, it's not like . . . I don't know how COVID will impact in the longer timeframe . . . I have learned to appreciate different things, that for example even though I have spent much more time alone and my number of close friends has degraded a lot, I know who the important people in my life are and who I really want to spend time with, and those are people I want to find time for daily, when . . . For example, you have an important hobby you can't do due to COVID, now you've seen the impact of that hobby on mental health and . . . Yeah, just like being able to concretely think about the things you value in life, but . . .

(Aino, 20-year-old student in Pizza Restaurant.)

According to Heidegger [46] solitude is not a negative condition to be avoided but an abiding human condition to be embraced [47]. Loneliness can provide a way for youth to confront their true selves and discover answers about the meaning of life. Loneliness is not only an ontological reality but also a psychological necessity for a coherent, authentic life. Luckily, young people have had social bonds and connections with each other through social media and work.

Finnish research on the polarization of young people [48] reported that those young people who experienced solitude when moving out of their childhood home were at risk of exclusion. The statistical data show, however, that there was no significant correlation between feelings of loneliness and living on one's own. Loneliness in that study was gender-biased. Young men experienced more loneliness than young women, who had more intensive friendships than young men.

In a life course approach, the meaning of context and circumstances becomes obvious. In my longitudinal study on the formation of Finnish young people's worldviews, the pathways of the subjects of the study linked them to childhood experiences, the timing of life course transitions, and associated adult outcomes capturing social exclusion or inclusion in their youth [37]. The longitudinal study of Ingrid Schoon [49] found evidence that many young people have the capacity to meet and overcome challenges and use them for growth or the maintenance of adaptive functioning.

### 5.1.1.3. Thoughts about Future

. . . When I was younger, getting closer to being an adult, I was dating for a longer time, and back then I was living more calmly and planning things in life more. I had different dreams, but now that I've lived here and have been free, that's now even further away. The five-year plan is, like, that I want to do a lot of things and go abroad, etc., but after the five years, I am over 30 years old, 31, 32, and at that point it would probably be the time to think about the next moves. But now it's not very present, and I am really excited about snowboarding, so if you don't have that as a hobby, you won't understand why someone must be there 150 days a year and go down the hills. And it's more difficult to fit, and I don't really want to compromise either, because everything seems good in life already.

(Lauri, 26-year-old in Ski Rental Company.)

The young people in this study had experienced more optimism than pessimism in their pandemic time in Lapland. Some of them explained stress about school, work, and unemployment, but in their mind, it was not so serious. Of course, the coronavirus pandemic has changed their normal life, but it has also given them more free time for activities and, for example, better accommodations than in normal times. They have had time to think about their life and plan the future more realistically.

. . . well, last year when I came back from here, I was supposed to have work, etc., but all of that went away because of COVID, and I didn't really find work,

so I was unemployed last summer because of COVID. It does change [the plans for the future HH]. . . . Yeah, I was living with my parents. Here, I started off in a communal cabin, and one positive thing COVID has brought to me was that I was moved to an individual one, so I got a new cabin. I got a better cabin because of it. There is one positive thing. . . . Well, I have always been that type of person, that I have enjoyed sitting down, or just being an introvert, so I must admit that when there's a lot of people, it feels like the walls are closing in and I can't deal with it, I am too used to living like this, where no . . . Quarantine, the time, went with no problems. . . . I played games and watched movies and went outside a bit too.

(Aatu, 20-year-old in Ski Rental Shop,)

Well, as for the near future, I have decided to spend the summer in Helsinki, where all my friends and family are, so I won't go during the summer . . . again in the further future, this has made me think about my future career more realistically. That I feel like if it was a normal year, I am the type of person that would . . . I would have integrated into the student culture. . . . It would have pulled me in very strongly . . . which are things that you want to find time for in your daily life, because . . . Well, for example, if you have an important hobby that you couldn't do because of COVID, you have noticed its effect on mental health already, and . . . Yeah, just like that, have been able to concretely think about one's own morals, but . . .

(Aino, 20-year-old student in Pizza Restaurant.)

[thinking about the future, HH] . . . it's quite short-term, but I do have some kind of bigger picture in my mind, that there are different steps that you know or have some kind of picture where you want to be in 20 years, but it's very foggy and on the higher level. But then otherwise, I don't know. In my current work or other things, I don't think ahead very much; now the plan is to work the next half a year and look again in the fall . . . for example, talking about working life, thinking about income, I have some accommodation services here . . . [name removed, HH]. I don't know if I will live here myself at that point . . . but I want to, anyway, to have a place here in Finland, and in the upcoming five years I want to do as much downhill skiing as I can and go to the big mountains when I am still young and able, or at least think I am. I would like to challenge myself on that too. Then also balance it out, so that I also go ahead in my career, so that I don't completely stop one thing or another. But the business mindset has been very strong for me, especially in the past; now I have been here for a couple of years, but after upper secondary school, I had a business for around four or five years. For the most part, it is what I will be doing mostly. It's hard to imagine that someone would offer interesting and good careers, that I would jump into an office after this. So, here I have been very satisfied.

(Pekka, 27-year-old in Nature Center.)

Research in the U.S. by Shrikanth, Szpunar, and Szpunar [50] suggests that people generally think positively about their personal future and negatively about the future of their country. One possible concern regarding the future is the extent to which people think about the distant future. That is, do they really think about events that might take place after 5–10 years? The study of Shrikanth, Szpunar, and Szpunar showed that these personally positive and collectively negative thoughts have implications for how people think about the world. People were more likely to think of positive than negative consequences of social media and artificial intelligence when adopting a personal perspective, whereas they were more likely to think of negative than positive consequences when adopting a collective perspective. The possibility of thinking about the personal past as positive and the collective past as negative may give rise to more general biases that guide expectations of the future. This definition seems well suited to a young person's own interpretation of future worries and fears. The young people in Lapland were well aware of global

environmental issues, and those came up in all the interview discussions. Concerns were also raised about the sustainability of Lapland's environment if tourism grows year-round and the number of tourists grows. On the other hand, there were also concerns about the effects of global warming on Lapland's tourism. Climate change was seen as scary. It is also combined with the increase in diseases and pandemics, which means concern about one's own work in tourism. It was recognized that one could have some influence with one's own behavior, as well as with politics, which was hoped for more transparency.

> . . . of course, it's related to my career too, climate change, that is. Now I think anyone who is around the same age as I am thinks of it at some level and is also scared of it. And COVID goes hand in hand with it, that the pandemics like that will become more common, but . . . I would say climate change is a good umbrella term for my biggest fear. The future dystopian scenarios are part of it, which I keep thinking about. Maybe the fear of not being enough, or can I somehow take part and make a big enough impact to make an actual change . . . that's quite difficult too, because Finland is such a small country and the COVID situation has gone all right in Finland, but I don't know if that's because of the government actions or because we don't have as many residents spreading COVID as much. In my opinion, or the only thing I wish from the government, not that it's directly related to COVID but all decisions, is to have even more transparency in who makes the decisions and what the decisions are based on. And when there have been some conflicts in the decisions made by the government and . . . statements, you just want to know why they exist.
>
> (Ritva, 22-year-old in Sport Shop.)

Climate change is a global concern. The Finnish study by Ratinen and Uusiautti [51] showed that students had relatively high constructive hope rather than denial hope when it comes to climate change. Additionally, this hope was not built on the minimization of climate change. The young people interviewed acknowledged the fact that being optimistic about the influence of one's actions (or opportunities to contribute, for example, by avoiding flying) is an important part of hope. It was seen also as a positive impact of the COVID-19 pandemic because people have not traveled much during the pandemic.

> . . . I have been thinking about it a lot, that if and when the borders open, everyone is going to be really excited about traveling. I would like to go traveling, but I also would like to do it the way where there's the least possible amount of flying, for example. Or that you spend a long time in the one destination. I am a bit scared that now that the flights have been limited and their amounts reduced significantly, how will it go when there are more of them again, and overall, will the pointless wasteful consumption blow up again? I hope that it would still stay controlled . . . like last year, there were more things, like, about the water in Venetian canals, that the water was so clear that fish would swim in there again . . . or somewhere in China, they could see the sky again, because there's been so much air pollution, that now when the borders open, if they open, it'll make the Venetian canals go back to a horrible condition and the environment overall because of traveling, so yeah, but I mean, nothing really scares like that, but these are some of my worries.
>
> (Erkki, 21-year-old in Sport Corner.)

### 5.1.1.4. Concerns about Tourism in Lapland

Outdoor recreation and nature-based tourism within a socioecological system are related to global and regional economies and transnational regulations. The youths understood that they affect trade, business development, infrastructure and utility development, investment resources, and access to workers. Visitor preferences for outdoor activities in Lapland are often shaped by media images and consumer trends, which constantly shift the flow of visitation, the outdoor activities pursued, and the social patterns of travel. For ex-

ample, when certain destinations in Lapland catch on with social media, site managers may be unprepared for the onslaught of visitors. Societal trends in work and leisure patterns and practices also shape how much time visitors have available to explore the outdoors. For the interviewed young people, nature in Lapland was very important and they were concerned about ecological changes, which growing tourism can affect.

> The nature of Lapland . . . Well, it is a topic that is a bit worrying, where I personally, I see it this way, that it's nice that the travel industry is growing and we are trying to make it more year-round and so on, but we should focus on making sure it's built on a sustainable base. The difficulty is there, of course, that money matters a lot. And also, how far into the future we are looking.
>
> (Maija, 21-year-old in Sport Shop.)
>
> . . . Sometimes, I mean, there is enough space and people coming, and there's not really problems like that, but in the bigger picture I think we should think about that. It's very hard to look 40 years into the future, to see how it will be here at that point. I am very bad at stressing out or being scared, I mean, I don't take it personally or lose sleep thinking about these topics very easily. . . . I like that people have found Lapland and went there to go to the outdoors. . . . But for me, it feels like this winter there have been more people of my age than ever before downhill skiing and so on; a lot of people came here to work remotely. In a way, it's quite fun, that when everything is closed and we are trying to be in contact with people as little as possible, for me it has been quite the opposite. For me it has been the most social winter here. It's not that much, but usually you could've counted the number of people on one hand that you climb and ski down those hills with. Now, the group is twice that, maybe even three times in size. I feel like I have lived the way I wanted, and I feel like I am living the life.
>
> (Pekka, 27-year-old in Nature Center.)

### 5.1.2. Education and Work

A comparative study of university students' coronavirus experiences [43] showed that university students have been particularly affected by COVID-19. Remote teaching and social distancing measures implemented across institutions worldwide have dramatically changed campus life. The pandemic may affect future educational opportunities, job prospects, and financial stability [52]. Recent Finnish studies of COVID-19 have reported that students' study burnout can lead to depressive symptoms and increase the risk of dropping out from studying by four times, whereas study engagement can promote both life satisfaction and success in future educational transitions [53–55]. Those young people engaged in studies or work have the energy and will to put effort into their work; they feel driven, enthusiastic, and inspired. Study and work environments provide the opportunities and resources for engagement, and young people's skills, needs, and motivation determine how they engage with those opportunities [55,56]. In this study, those who were studying remotely from Lapland thought that the time was heavy for them, but it was also a good opportunity for them to have it a little easier. However, students in remote studies were worried about getting in touch with their coursemates and normal student culture.

> . . . The time studying has been very . . . Or I have felt it as very heavy, but I also understand that a lot of people have it much worse. I have tried to somehow outsource the effect of COVID, and I understand that studying has been difficult because of COVID, and remote studying and so on; I have tried to not put as much pressure on myself because of it. . . . I have been here now since the beginning of February, and that wouldn't have been possible for me without COVID, because I can now do some studies remotely from here, so . . . I don't have any motivation to return . . . and somehow the negativity caused by COVID is somehow associated with it. . . . In the fall when the studies started, it was . . . you could organize some events, so you were able to integrate into the culture

and socialize with the people at least a bit. And in that way, we have been lucky that we have very few students in our program, so we have been able to meet up with quite small groups, but during the winter . . . and there's nothing in common . . . [everyone] studies remotely, so there is nothing like hanging out after the lecture, or going out for lunch, or . . . It requires a lot of initiative to keep in touch with the other students and . . . there hasn't been time to form connections with the other people.

(Aino, 21-year-old in Pizza restaurant.)

Those who had a degree in tourism had thought about further studies, for example, a master's degree, abroad. Studying in a foreign language was considered challenging. However, according to the interviewees, COVID-19 had not changed their own plans.

. . . that it would be nice to study something. I have had that option in my mind, that I would go back to school, but I am not in a hurry with it. More I would like to do it at the point when I actually find something I am truly interested in.

(Laura, 20-year-old in Ski Rental.)

. . . Would it be a master's degree? Possibly. In that I had realized that originally, I had thought of studying traveling first, and then I wanted to graduate as a Master of Marketing, because this is the field where I am, traveling marketing, but then I realized that with a Bachelor of Hospitality Management, I can't apply directly for a master's degree, so since then I haven't been motivated to study, but I don't know about the future. . . . I have always been bad at languages. It has always been a challenge for me. So, it's funny that I am in the travel industry. Somehow, I end up with my own challenges in life. I have been thinking that I want to live abroad for a moment, because I have never taken part in a student exchange program or been abroad for a longer period of time, where I would need to survive in a foreign culture with a foreign language, so it would be interesting to take myself to experience some discomfort in that. Normally it's not like, it feels like, that for a lot of people it would be just really cool to go abroad, and especially for younger people it feels like that everything is wrong in Finland and abroad things are better. I have never thought that way; I think it's amazing to live and be here. But then at some point in my life, I wanted during the next five years to spend more time abroad and challenge myself . . . But of course, you can't, like . . . You don't know what COVID's effect will be in the long term. I hope and believe that in the end it won't be very much, so the travel restrictions will be lifted and so on. I haven't . . . Well, at least my plans for the future haven't changed a lot—at least for now.

(Pekka, 27-year-old in Nature Center.)

The COVID-19 pandemic in Lapland might foster a range of side skills, such as autonomous thinking, and enable different social contacts and interactions, offer chances to build networks, establish new friendships, and provide experiences of belonging to the working life in tourism. The pandemic might represent an important developmental context for young people to acquire their developmental tasks, unfold their potential, and experience a sense of belonging in their peer group during work and leisure time. The other option is that the COVID-19 crisis is challenging young people's engagement too strongly and may hinder their abilities to thrive in their future education and work life. Longitudinal studies have shown that engagement in education and work predicts successful educational and work transitions and later satisfaction with chosen educational and work-life pathways [54,55].

## 6. Discussion

*COVID-19 as a Transitional Time*

Youth is a period of transition from childhood to adulthood. In transitions of young people, their institutions are in education systems and labor markets, but their experiential spaces are mostly in peers, fun-and-games groups, and social media. On their path to adulthood, they are confronted with many institutional demands of education and work. In the sociological life course theory, the timing and direction are embedded in structural conditions including social background, educational and welfare systems, and the labor market [52,57]. In social-psychological theories, these institutions present challenges for youth identity formation. Young people are embedded in their life perspective, identity, individual capacities and competencies (agency), motivation, and life goals [8,58–61].

The lives of the young interviewees in Lapland were in a transitional period. They were only there working a short time. After leaving Lapland, they must decide what to do next after COVID-19 and their short-term jobs. They had to think about their life further after their COVID-19 experiences. Young adults are in a period in their lives in which they must plan their future and make significant decisions regarding many aspects of their adult life [24,38,58,60–63]. During emerging adulthood, their expectations for the future are especially important and could influence their goal-setting and motivation to accomplish those goals.

Some of these young people found their individualization process in Lapland liberating, freeing them from normal student and work-life norms. Some had inherited structural advantages such as their own cottages in Lapland. In making transitions, they may have had more family and economic resources to ride out the educational delays that are commonly encountered now during the COVID-19 pandemic. They may also have drawn on their parents' networks to eventually find a workplace. This "freedom" in Lappish nature, skiing, and other leisure activities can come with a price for those who cannot capitalize on it, either because of economic disadvantage or because they lack the type of agency required to be "architects of their own destiny" [61,64]. These people face the "individualization contradiction" [59]. In the late-modern world, young people are expected to be the architects of their own destinies and to shape their own "choice biographies" [65,66], but for many young people, the avenues do not exist to turn this expectation into a reality. For some, choices are unviable because their economic circumstances are unfavorable for engaging in the many years required to reach occupational destinations.

A study about young, seasonal workers in Lapland ten years ago showed that many young people did not want to commit to any worldview or even a romantic relationship [8]. The reluctance to participate in work and the value of the quality of life, which includes experiencing new things, were more important than the standard of living. Work was not an explanation that defines one's identity, but a young person can build his or her identity through leisure activities and hobbies. Part-time jobs are part of the daily lives of many young adults in their transitional phase in COVID-19 times, and society seems to support it. For example, EU projects and many academic ones are short-term projects. Many young people consciously apply for part-time work, and at least they are not bothered by the short duration of work. This was the case also among the interviewed seasonal workers in Lapland. They valued the work, which allowed them a wide range of independence and freedom.

We turn to probably the biggest challenge of our time, the destructive force of the COVID-19 pandemic, which in the long term may turn out to have the most significant implications for young people [52]. Youth conditions with different transformations and prerogatives are undoubtedly influenced by the social circumstances surrounding them. The literature describes a world of young people's trajectories that are often unstable, risky, individualized, pluralistic, and accelerated [11,64,67–69]. The paths into adulthood do not diminish the long-lasting influence of childhood family status, gender, and ethnic origins [37,70–72].

This hermeneutic-phenomenological study shows that the COVID-19 period in Lapland gave the interviewees freedom from a normal commitment to study and from family

life. It gave them time to think about their own choices that will affect their future. There was also time to reflect on their own values (cf. psychosocial moratorium by E.H. Erikson, [73]). They were able to actively pursue things that were important to them. Many of them appreciated hiking in nature and winter sports. Indeed, they considered themselves to be in a privileged position in Lapland. However, we have to keep in mind that young people need political support in their transitions to adult citizens so that their life courses are not dramatically affected by events such as the COVID-19 pandemic and other crises that take place during their youth. Understanding why this is the case is of critical importance to young people's futures as well as society in general.

Despite the ongoing COVID-19 pandemic crisis, the interviewed young people seemed to have positive expectations about their future. Temporary work provided them chances to learn and practice skills they might need in the next job. It also gave them a forum for identity development and opportunities to build new social ties and social capital. Their hobbies and other daily activities connected their value worlds to work and leisure time, which in turn influence and are influenced by the multi-layered ecology within which their lives are embedded from the proximal contexts of everyday life. Human resources, capital resources, and technology dictate a considerable amount of the normal lifestyle of youth. Because of the COVID-19 pandemic, their lives were not normal, and in these exceptional and uncertain transitions, the young person's education level, understanding, and welfare dictated what information he or she had about the school-to-work transition.

The transition to adulthood means for a young person that he or she is facing major decisions whose effects may last a lifetime. The young interviewees were wondering if they should study more or start a job, maybe move somewhere else or even to a foreign country to study, or would it be wiser to stay in the place where they had lived? Should they get more education, and what could it be about? Such decisions may substantially impact a young person's life. They may be "life-changing" as, for example, the COVID-19 pandemic is a new situation that may change young people's decisions about the future. The COVID-19 period can change the future horizons with completely new experiences, which may have impacted the way of valuing certain things in life. It is related to the sense of identity in the sense of a personality as well as the notion of identity in the sense of sameness [73–77].

## 7. Summary

In this qualitative study, I interpreted young people's perceptions and orientations from a hermeneutic-phenomenological perspective. Hermeneutics and phenomenology are philosophical trends that have their own starting points and traditions, which are not very well known in youth research. The methodology is very laborious. I chose this research framework for a possible in-depth understanding of coronavirus experiences and meanings among young people in the education and labor market in Lapland.

In a discussion-type semi-thematic interview, it was possible to delve into coronavirus experience topics, and the interviewees could bring up coronavirus narratives during the interview on their own initiative. As a data collection method, the interview became a conversational face-to-face interaction situation, in which the interviewee and the interviewer interacted, and COVID-19 rules were followed.

Interviews on ten young people's coronavirus experiences and their implications for the transitions from education to employment and future orientations were partly discussions of topics related to education, work, and the transition to adulthood combined with young people's COVID-19 experiences and their implications. In the interviews, young people combined their life experiences and perceptions of the world with the coronavirus experiences. The COVID-19 interpretations of young people had positive and negative meanings for their transitions in education and the labor market. Despite the ongoing COVID-19 pandemic crisis, the interviewed young people seemed to have positive expectations about their future. Temporary work provided them chances to learn and practice skills they might need in their next job. Their hobbies and other daily activities

were connected to their value worlds, to work and leisure time, which in turn were influenced by their lives in contexts of everyday life. Even though they were concerned about the pandemic, the health of their loved ones, and they felt loneliness, they were satisfied with their lives in Lapland and thought about their personal futures positively.

One of the challenges of analysis was data compression. It is not easy to preserve the diversity of the material and find the similarities and differences in the whole data. The interpretation and analysis process is such a dynamic circle, looking at individual interviews and the whole data in turn.

The aim of this article has been to bring young people's voices to the fore. Objectivity in hermeneutic research can be understood as openness to the research text. This qualitative research is not intended to generalize observations but to provide an empirical description of the perspectives (horizons) of the interviewed young people. Reflexivity is an essential part of strengthening the reliability of the research. This includes a critical examination and questioning of the researcher's own preconceptions and attitudes. [78,79]. In this, the seminars and discussions with my COVID-19 research colleagues helped me.

When the young interviewees interpreted their own experiences, observations, and thoughts in speech, they also formed self-understanding and outlined what kind of future and life they wanted to live. The young, interviewed persons in seasonal work during COVID-19 in Lapland saw their own lives with optimism.

> Now is the best time to be here. Many other friends think that now I've got to finish school, then work, and then something else, and that now, I've just got to work. And personally, I find it funny, that life is just amazing all the time.
>
> (Lauri, 26-year-old in Ski Rental Company.)

**Funding:** This research received some funding from Faculty of Social Sciences at Tampere University.

**Institutional Review Board Statement:** Not applicable.

**Informed Consent Statement:** Informed consent was obtained from all subjects involved in the study.

**Data Availability Statement:** The interview data will be saved at open access Finnish Social Science Data Archive https://www.fsd.tuni.fi/en/ (accessed on 17 May 2022).

**Conflicts of Interest:** The author declare no conflict of interest.

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
