# Peer review of "Hermeneutic-Phenomenological Interpretation of Coronavirus Experiences, Their Meanings, and the Prospects of Young Finns in Education and the Labor Market in Lapland"

_societies, doi:10.3390/soc12040112_

Round 1

Reviewer 1 Report

The manuscript “Hermeneutic-phenomenological interpretation of the coronavirus experiences, their meanings, and prospects of young Finns in education and the labor market in Lapland”, is a very interesting and original piece of research on young people’s lives during pandemic times, precisely young workers in Lapland during confinement times.

The results show original and innovative conclusions that will contribute to the scholarship on youth studies, relating the meanings of coronavirus experiences with structural and biographical processes young people are under thru.  

The manuscript is very well-grounded in theory and methodology, as well as in empirical data. However, an effort should be made by the authors to turn clearer and stronger a theoretical argument for the paper, that conducts the narrative of the demonstration, summary, and discussion of research findings. I found the structure of that narrative very random, without a strong and clear argument driving. The “Research findings” part sounds more like an empirical report than an academic article. The authors can also be more restricted in the references used, using only the ones that give strength to the main argument. A shorter title closed to the argument could be also a strategy to make it clearer for the reader. 

I would also suggest to the authors introduce a more analytical perspective in the approach to the research findings, and not present them so raw. This will give clarity and strength to the argument. To give some socio-demographics about the interviewees’ manuscript give “voice” – as is the case of the voice used as an epigraph for the manuscript, from who we know is a 21-year-old student, applying for art education; gender should be considered as well – is also advised.

After that formal revision of the structure and narrative of the manuscript, I think the article will be in condition to be published in Societies, with great academic success.

Author Response

Thank you, Reviewer 1, for your valuable feedback about my article manuscript “Hermeneutic-phenomenological interpretation of the coronavirus experiences, their meanings, and prospects of young Finns in education and the labor market in Lapland”.  

I have made changes and added two Figures to turn clearer and stronger theoretical argument for the paper from the hermeneutic-phenomenological framework. I have updated the discussion chapter linking the research themes discussed with young people  (young people´s horizons) to the research literature. Discussion chapter is horizons fusion. Summary has also added to the article. I hope that the figure 2 helps to understand the structure of the narrative and the argument driving. The “Research findings” are interpretation of young interviews horizons with my understanding and synthesis. The aim is that the voice of interviewed young people is heard in findings. I have tried to check my references more restricted. I also added shorter titles to make the arguments clearer for the reader.

The voices of young people are distinguished by pseudonym name, age and by the place where the speaker worked. I hope that made revisions in the structure and narrative of the manuscript will be now accepted for publish in Societies.

Thank you very much for your helpful advice.

Reviewer 2 Report

Overall, this paper is more descriptive than analytical. It has quite a small data set and spends much of the paper presenting large verbatim quotes. Whilst it is claimed that the author: ‘interpret(s) young people’s perceptions and orientations from a hermeneutic phenomenological perspective’ – there is insufficient evidence of this, and it is not clear how this has been done or what the implications of this analysis has been on presentation of the data or discussions there of… (Also I’m not sure it is the case that hermeneutics are not well known in youth studies – and this claim is certainly not evidenced)

It is possible that a significant contribution is being made in this paper – but this is not evidenced or presented in a way that a judgement can be made upon it yet... Not least because some of the findings presented are extraneous to the central research question of the impact of the pandemic – such as climate change and the reality of rural life in Lapland. A much more coherent argument needs to be established. For example, on line 756 the author suggests that: ‘The pandemic might represent an important developmental context for young people to acquire their developmental tasks and unfold their potentials, and to experience a sense of belonging in their peer group during work and leisure time. [whilst the author also considers] The other option is that the COVID-19 crisis is challenging young people’s engagement too strongly and may hinder their abilities to thrive in their future education and work life’. The author needs to interpret their data in the light of this option . What is it showing…? Which way is ‘this’ data leaning…?

Overall, the primary finding seems to be (from line 763) that: ‘the interviewees freedom from a normal commitment to study and from family life. It gave them time to think about their own choices that will affect their future. There was also time to reflect on their own values – is that all.  This is in itself interesting, and could be significant but it is not clear whether it is because it is not presented in the context of other research. (See point later about literature review and the wider academic context of this paper.)

The following sections (772- 791) do also contain some interesting  potential discussion points but these need to be integrated more fully with the data. Overall there is too little integration of the data, the literature and the conclusions. The overall argument needs to be much tighter.

Research findings:

In the findings section the data is rather poorly presented with a series of very large ( anonymous) quotes to illustrate a section but the quotes do not always match the previous ( or subsequent discussions of the literature) some quotes don’t appear (directly) related to Covid either- eg see quote on line 340. At other times the discussion does not relate to the findings at all e.g. The reference and discussion to gender does not link to any data  - nor does loneliness it appears. It would also help if the quotes were attributed to each participant by (pseudonym or a number with gender and age

The paper needs to have a clearer focus on Covid - there is not enough that relates directly to YP’s experience of Covid – and there is little reference to any of the emerging literature on the impact of the pandemic on young people (see end note for examples[i] ) other than the brief discussion on lines 741 – 748. The paper would be better constructed if there was a literature review of current research on the impact of covid on young people. Then the author can ascertain whether their findings are consistent with previous research.

Summary and discussion

This section also has weaknesses – there are some sweeping generalisations made – drawing generalising conclusions about all young people from a sample of 10 interviews – eg – ‘I have noticed a remarkable change in the future orientations of today’s young people (line 709) and even the young people of the 1990s’.  Also line 719 the author concludes that:  It seems that many of today’s young people live in the present moment’ it is not at all clear where this conclusion is derived from or how data from 10 young people can support that conclusion…?

There is also no overall conclusion to the paper

Further comments

On Page 8 the author twice mentions their reflections on their own position but doesn’t say what is the impact of this…? Or how their approach altered as a result…

Page 9 Individual experiences and collective world view section is a bit vague it is not clear what the purpose of this is ?

‘Gotten it’ is an example of American slang see line 244 – there are others too, these need to be removed

Line 481 the contrast between the yp who ‘floundered’ and those who didn’t is not evidenced.

·         Cunning, C. & Hodes, M. (2022) The COVID-19 pandemic and obsessive-compulsive disorder in young people: Systematic review Clinical Child Psychology & Psychiatry, 2022 Jan;27(1):18-34. DOI: 10.1177/13591045211028169

·         Haig-Ferguson, K. Cooper, E. Cartwright, M.E. Loades and J. Daniels (2021) Practitioner review: health anxiety in children and young people in the context of the COVID-19 pandemic in Behavioural and Cognitive Psychotherapy Volume 49 Issue 2 , March 2021 , pp. 129 – 143 DOI: https://doi.org/10.1017/S1352465820000636

·         Danese, A. & Smith, P. (2020) Debate: Recognising and responding to the mental health needs of young people in the era of COVID-19 Child and Adolescent mental health Volume25, Issue3 Pages 169-170 https://doi.org/10.1111/camh.12414

Author Response

Thank you very much for your helpful advice.

I have considered the suggestions, thanks for them. I have made changes and added two Figures to turn clearer and Stronger Theoretical argument for the paper from the hermeneutic-phenomenological framework. I hope that the figure 2 helps to understand the structure of the narrative and the argument driving. The "Research findings" are interpretation of young interviews horizons with my understanding and synthesis. I have updated the discussion chapter linking the research themes discussed with young people (young people's horizons) to the research literature. Discussion chapter is horizons Fusion (see Figure 2). The aim of this article has been to bring young people's Voices to be heard as their focus is on COVID experiences and meanings. Objectivity in hermeneutic research can be understood as openness to the research text. In this qualitative research it is not intended to generalize observations, but to provide like in case study an empirical description of the perspectives (horizons) of interviewed young people. I hope that the figure 2 helps to understand the structure of the narrative and the argument driving. The "Research findings" are interpretation of young interviews horizons with my understanding and synthesis. I hope that there is now sufficient evidence about how the research has been done and what are the implications of the analysis.

I hope that the changes have established more coherent and tight arguments. The revised discussion chapter is the Fusion of Horizons linking the research themes discussed with young people (young people's horizons) to the research literature. Summary has also been added to the article. The aim is that the voice of interviewed young people is heard in the findings. I also added shorter titles to make the arguments clearer for the reader. The Voices of young people are distinguished by pseudonym name, age and by the place where the speaker worked.

I have added a literature review of current research on the impact of covid on young people. Thank you for your proposal. I hope that the paper is now better constructed

I have cut the texts 'I have noticed a remarkable change in the future orientations of today's young people (line 709) and even the young people of the 1990s'. and on line 719 the author concludes that: 'It seems that many of today's young people live in the present moment'.

I have changed the text in Page 8 likewise I have cut the page 9 Individual experiences and collective world view section.

'Gotten it' has been corrected and I hope that others too.

Thank you for the references!

I hope that revisions made in the manuscript will now be accepted for publication in Societies.